# Effective Adaptation in Multi-Task Co-Training for Unified Autonomous Driving

**Xiwen Liang**[1*], **Yangxin Wu**[1*], **Jianhua Han**[2], **Hang Xu**[2], **Chunjing Xu**[2], **Xiaodan Liang**[1†]

[1]Shenzhen Campus of Sun Yat-Sen University, [2]Huawei Noah's Ark Lab

{liangxw29@mail2, wuyx29@mail2, liangxd9@mail}.sysu.edu.cn,
{hanjianhua4, xu.hang, xuchunjing}@huawei.com

## Abstract

Aiming towards a holistic understanding of multiple downstream tasks simultaneously, there is a need for extracting features with better transferability. Though many latest self-supervised pre-training methods have achieved impressive performance on various vision tasks under the prevailing *pretrain-finetune* paradigm, their generalization capacity to multi-task learning scenarios is yet to be explored. In this paper, we extensively investigate the transfer performance of various types of self-supervised methods, e.g., MoCo and SimCLR, on three downstream tasks, including semantic segmentation, drivable area segmentation, and traffic object detection, on the large-scale driving dataset BDD100K. We surprisingly find that their performances are sub-optimal or even lag far behind the single-task baseline, which may be due to the distinctions of training objectives and architectural design lied in the *pretrain-finetune* paradigm. To overcome this dilemma as well as avoid redesigning the resource-intensive pre-training stage, we propose a simple yet effective *pretrain-adapt-finetune* paradigm for general multi-task training, where the off-the-shelf pretrained models can be effectively adapted without increasing the training overhead. During the *adapt* stage, we utilize learnable multi-scale adapters to dynamically adjust the pretrained model weights supervised by multi-task objectives while leaving the pretrained knowledge untouched. Furthermore, we regard the vision-language pre-training model CLIP as a strong complement to the *pretrain-adapt-finetune* paradigm and propose a novel adapter named LV-Adapter, which incorporates language priors in the multi-task model via task-specific prompting and alignment between visual and textual features. Our experiments demonstrate that the *adapt* stage significantly improves the overall performance of those off-the-shelf pretrained models and the contextual features generated by LV-Adapter are of general benefits for downstream tasks.

## 1 Introduction

Multi-task learning is a long-studied problem in computer vision and has become an emerging paradigm in autonomous driving [19, 50, 18]. An autonomous vehicle may need to concurrently perform a wide range of perception tasks, e.g., locating the pedestrians and cars, deciding road affordability for driving, as well as detecting the lanes to precipitate a driving action. The joint learning of multiple tasks can not only reduce the training and inference time, but also act as a regularizer that enforces the learning of generalizable representations [6, 19]. Apart from efficiency and simplicity, an all-in-one architecture is argued to have the potential to enhance the robustness of the driving system by implicitly learning the synergy between heterogeneous tasks [19, 45, 40].

---

[*]Equal contribution.
[†]Corresponding author.

36th Conference on Neural Information Processing Systems (NeurIPS 2022).

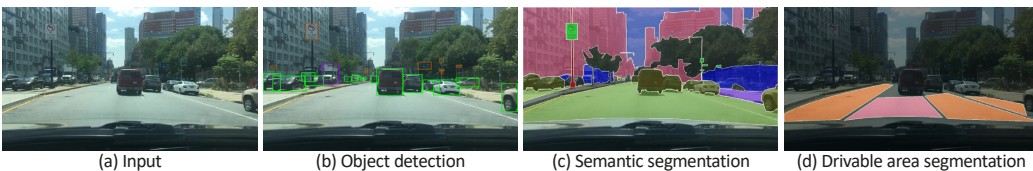

| (a) Input | (b) Object detection | (c) Semantic segmentation | (d) Drivable area segmentation |

Figure 1: Our multi-task model takes (a) an RGB image as input and tackles (b) traffic object detection, (c) semantic segmentation, and (d) drivable area segmentation simultaneously.

In the pursuit of the merits of multi-task learning, it is crucial to obtain universal features with great transferability. Recently, many concurrent self-supervised pre-training methods have shown great potential when transferred to different types of vision tasks [47, 49, 44] under the *pretrain-finetune* paradigm. Despite the impressive performance, their transferability to multi-task learning scenarios is yet to be explored. We argue that the heterogeneity of tasks will introduce a multitude of challenges to train a unified model and it is often not the case that multi-task learning is of universal benefit. Firstly, the widely adopted *pretrain-finetune* paradigm may lead to degraded performance in multi-task learning due to the misalignment of objectives between pre-training and fine-tuning [12, 16] since most supervised and self-supervised methods are highly specialized for a specific type of objective or task [3, 4, 47, 49]. Secondly, the performance of multi-task learning relies on many non-trivial factors including model architectures, data augmentations, hyperparameters, convergence properties, etc [36, 39]. The specialized techniques catered for a specific type of architecture or task are largely not applicable to a universal architecture since they are prone to fail during generalization. Moreover, due to the labor-intensive process of data annotation, it is hard to collect complete annotations of different granularities for all tasks, which further complicates the situation.

In this paper, we focus on heterogeneous multi-task learning on partially labeled data (HMPL) under the realistic scenario of autonomous driving. We first delve into the representation learning stage of HMPL to reveal the performance degradation of different pre-training methods. We thoroughly examine a wide range of pre-training methods including supervised pre-training (pretrained on ImageNet [33] ), classification-oriented methods (e.g., SimCLR [3], MoCo [15]), detection-oriented methods (e.g., DetCo [47]), segmentation-oriented methods (e.g., DenseCL [44]), and vision-language pre-training methods (e.g., CLIP [29]) on three fine-grained tasks on the large-scale driving dataset BDD100K [52], i.e., object detection, semantic segmentation, and drivable area segmentation. Surprisingly, the performance of these methods varies greatly under the same training protocol, especially on the dense prediction task. For example, MoCov2 [4] only gets 12.5 mIoU in semantic segmentation and segmentation-oriented models like DenseCL [44] and DetCo [47] perform unsatisfactorily on dense prediction tasks, which suggests that the misalignment lied in the *pretrain-finetune* paradigm can lead to prominent performance degradation and there exists much room for improvement.

Given that 'universal' pre-training is still unsolved and redesigning the resource-intensive pre-training scheme comes with great computation overhead, we aim to develop a general approach that can fully harness the knowledge from the off-the-shelf pretrained models and make them amenable to multi-task scenarios via efficient adaptation. We draw inspiration from the recent progress of prompt-based learning in NLP [24, 55, 29, 10], where the language model pretrained on massive amounts of raw text can be adapted to new scenarios with high efficiency by introducing hand-crafted or learnable prompts. Following this philosophy, we propose a simple yet effective *pretrain-adapt-finetune* paradigm for multi-task transfer learning as a substitute for the dominating *pretrain-finetune* paradigm in computer vision. Concretely, during the *adapt* stage, learnable multi-scale adapters with a small amount of parameters are tuned with frozen random initialized task-specific heads to dynamically adapt the knowledge from the pretrained models under the multi-task objectives. Albeit simple, we show that the *adapt* stage mitigates the gap between pre-training and fine-tuning and significantly improves the overall performance across different pretrained models while being very effective and introducing no extra training overhead.

Apart from the supervised and self-supervised pre-training, Contrastive Language-Image Pre-training (CLIP) [29] manages to learn high-quality visual representation from an enormous amount of noisy image-text pairs. CLIP achieves breakthrough performance in zero-shot image recognition, which indicates that it might have great potential for improving the generality and robustness of multi-task learning. To this end, we propose LV-Adapter to further excavate the linguistic knowledge from CLIP

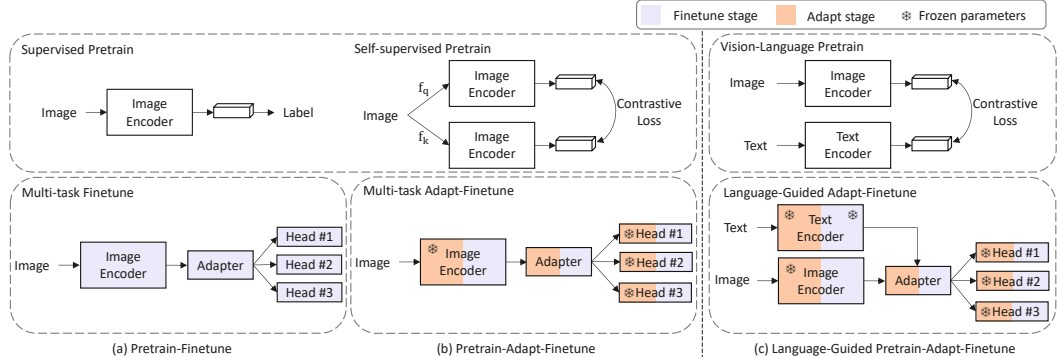

Figure 2: Comparisons of the conventional *pretrain-finetune* paradigm and our proposed *pretrain-adapt-finetune* paradigm. The language-guided *pretrain-adapt-finetune* paradigm further incorporates language priors into multiple downstream tasks.

and enhance the visual representations in the multi-task scenarios. To maximize the correspondence between the dense features and the class-specific concepts, we first learn task-specific prompts in an end-to-end manner. Then we model the language-to-vision adaptation function based on the cross-attention mechanism [43] to incorporate language priors in visual features.

We experimentally verify that our proposed *pretrain-adapt-finetune* paradigm significantly improves the overall performance of different off-the-shelf pre-training methods. For example, it obtains an absolute improvement by over 40% in semantic segmentation on MoCo-v1/v2, and DenseCL [15, 4, 44], and its superior performance does not rely on extra training costs or sophisticated architectural design. Furthermore, we showcase that our proposed LV-Adapter can serve as an effective complementary approach for multi-task learning. Especially, we experiment on three settings that correspond to different levels of annotation quantity and observe that LV-Adapter brings consistent performance gains on three heterogeneous tasks simultaneously. We hope our work can shed light on the heterogeneous multi-task learning by showing that effective adaptation is of great essence to mitigate the deficiency in the dominating *pretrain-finetune* paradigm.

## 2   Related Work

**Multi-Task Learning** Multi-task learning aims to improve efficiency and accuracy through shared information among multiple tasks. MultiNet [41] joints classification, detection and semantic segmentation via a unified architecture. Eigen *et al.* [8] address depth prediction, surface normal estimation, and semantic labeling. YOLOP [45] leverages CSPDarknet as the backbone and adopts three specific decoders to solve object detection, drivable area segmentation, and lane detection respectively. Standley *et al.* [36] and Fifty *et al.* [9] propose to identify proper task groupings for multi-task training instead of naively training all tasks together. In this paper, we focus on developing general and effective approaches for multi-task transfer learning in autonomous driving scenarios. We provide more discussions with several related works in the Appendix.

**Pre-training Methods** A dominating paradigm in computer vision is to pretrain on a large scale of data, e.g., ImageNet [33], then finetune on target tasks with usually less training data. Recently, researchers are interested in learning visual representations without human supervision. We roughly divided them into three classes, namely, classification-oriented, detection-oriented, and segmentation-oriented methods. Classification-oriented methods typically rely on contrastive learning and online clustering, e.g., SimCLR [3], MoCo [15, 4], and BYOL [14]. Detection-oriented methods like DetCo [47] are specially designed for object detection by conducting contrastive learning between the global image and local patches. Segmentation-oriented methods like PixPro [49] further work on pixel-level correspondence to benefit dense prediction downstream tasks. We show that these methods are sub-optimal for multi-task learning, and we focus on improving the performance of these off-the-shelf methods instead of redesigning the computation-intensive pre-training stage.

**Prompting Learning** Natural language prompting freezes large-scale pretrained language models and reformulates text input with example or learnable prompts to bridge the gap between pre-training

Table 1: Comparisons of popular multi-task training methods and our proposed LV-Adapter under the Disjoint-normal setting.

| Method | mIoU (SS) | mIoU (DA) | mAP | AP50 | AP75 |
|---|---|---|---|---|---|
| Zeroing loss [46] | 59.4 | 83.3 | 23.0 | 46.0 | 19.8 |
| Uniform sampler [22] | 59.9 | 83.2 | 24.3 | 47.0 | 21.8 |
| Weighted sampler [22] | 59.8 | 83.2 | 24.2 | 46.9 | 21.4 |
| Round-robin [22] | 60.7 | 83.1 | 24.2 | 47.0 | 21.5 |
| Self-training [12] | 60.3 | 83.1 | 24.9 | 48.1 | 22.2 |
| **LV-Adapter (Ours)** | **62.2** | **83.7** | **26.4** | **50.5** | **23.7** |

and model tuning efficiently. GPT-3 [1] first introduces in-context learning, prepending hard text prompts for every task before the input. Then many works based on such discrete text prompts have emerged, i.e., designing prompts manually [37, 34] and searching trigger words [35, 11]. Furthermore, recent methods inject learnable soft prompts [21, 26, 53] into the embedding space of the model and achieve impressive performance. In the field of computer vision, CLIP [29] trains on millions of vision-language pairs and demonstrates effective zero-shot classification. CoOp [55], CPT [51], and CLIP-Adapter [10] further tunes fixed CLIP with soft prompts by few-shot supervisions.

## 3 Empirical Study

### 3.1 Heterogeneous Multi-Task Setup

We concentrate on three challenging heterogeneous tasks in autonomous driving, i.e., traffic object detection, semantic segmentation, and drivable area segmentation, on large-scale driving dataset BDD100K [52]. We showcase the input and output of our model in Figure 1. We select the state-of-the-art detector and segmentation decoder head to build up our model. To be specific, we follow the hard-parameter sharing scheme [6] where each task shares the same backbone and has its task-specific head. We outline our model architecture in Figure 2 and elaborate each component in the following.

**Backbone & Neck.** We adopt the classic ResNet [17] as backbone and FPN [23] as neck to generate multi-scale features. The FPN neck [23] constructs pyramidal features, namely, $\{P_2, P_3, P_4, P_5\}$, which have strides of $\{4, 8, 16, 32\}$ pixels w.r.t the input image and fixed 256 channels.

**Segmentation Head.** We implement the decoder for segmentation based on MaskFormer [5]. MaskFormer formulates the segmentation task as a mask classification problem and its head is composed of a fully convolutional pixel decoder and a transformer decoder module. The pixel decoder takes $\{P_2, P_3, P_4, P_5\}$ as input and gradually upsamples the features to produce high resolution per-pixel embeddings $\varepsilon_{\text{pixel}}$. The transformer decoder module utilizes a fixed set of queries to attend to image features and produces mask embeddings $\varepsilon_{\text{mask}}$. Then $\varepsilon_{\text{pixel}}$ and $\varepsilon_{\text{mask}}$ is multiplied together to generate prediction masks. We refer the reader to [5] for more details.

**Detection Head.** We implement our detector based on Sparse R-CNN [38]. The detection pipeline constructs a fixed set of learnable proposal boxes (e.g., 300) which serve as region proposals and are fed into a series of heads for prediction. The dynamic instance interactive head dynamically generates the weights of a convolution filter that attends to the bins in the region proposals and predicts the location and category of object in a cascade manner.

We introduce three settings, i.e., the Disjoint-normal, Disjoint-balance, and Full-setting, corresponding to different levels of annotation quantity in our experiments. For clarity, we defer the detailed descriptions of three settings to Section 5.1. For consistency, we conduct evaluations on the same validation set of BDD100K for different settings above. We mainly work under the Disjoint-normal setting for comparisons between the performance of different pre-training methods as well as ablation studies. And we further verify the efficacy of our proposed methods on the Disjoint-balance and the Full-setting in Section 5.2.

Table 2: Comparisons of different paradigms under the Disjoint-normal setting with ResNet-50 backbone. Orange color indicates the results of our proposed *pretrain-adapt-finetune* paradigm, while others are results of conventional *pretrain-finetune* paradigm.

| | | Semantic Seg. | | Drivable Seg. | | Object Detection | | |
|---|---|---|---|---|---|---|---|---|
| Type | Model | mIoU | pACC | mIoU | pACC | mAP | AP50 | AP75 |
| Classification-oriented | MoCo-v1 [15] | 17.8 | 48.6 | 70.8 | 92.0 | 25.8 | 49.5 | 23.1 |
| | | 59.2 | 93.2 | 83.6 | 96.9 | 25.9 | 50.0 | 23.0 |
| | MoCo-v2 [4] | 10.3 | 19.8 | 73.4 | 93.5 | 26.0 | 50.1 | 23.2 |
| | | 61.2 | 93.4 | 83.8 | 96.9 | 26.1 | 50.4 | 23.4 |
| | SimCLR [3] | 60.3 | 93.3 | 83.5 | 96.8 | 25.4 | 48.9 | 22.5 |
| | | 60.1 | 93.2 | 83.5 | 96.8 | 25.2 | 48.9 | 22.3 |
| | SwAV [2] | 45.9 | 71.1 | 82.0 | 96.5 | 25.6 | 49.1 | 23.0 |
| | | 61.1 | 93.3 | 83.1 | 96.7 | 25.6 | 49.3 | 23.1 |
| | BYOL [14] | 59.2 | 90.2 | 75.6 | 93.9 | 25.9 | 49.8 | 23.4 |
| | | 61.7 | 93.4 | 83.5 | 96.8 | 25.7 | 49.4 | 23.1 |
| Detection-oriented | DetCo [47] | 38.1 | 58.5 | 83.2 | 96.7 | 25.9 | 49.7 | 22.9 |
| | | 61.0 | 93.4 | 83.7 | 96.9 | 26.2 | 50.3 | 23.3 |
| Segmentation-oriented | DenseCL [44] | 20.0 | 40.0 | 73.7 | 93.7 | 26.1 | 50.3 | 23.5 |
| | | 60.7 | 93.3 | 83.9 | 96.9 | 26.3 | 50.3 | 23.7 |
| Vision-language | CLIP [29] | 54.5 | 91.1 | 74.1 | 93.1 | 26.5 | 50.7 | 23.8 |
| | | 61.0 | 93.2 | 83.4 | 96.8 | 26.3 | 50.5 | 23.5 |

## 3.2 Multi-Task Learning Baselines

**Self-training** Based on the available annotations for each task, we trained three single-task teacher models on the labeled data and use them to generate pseudo labels on unlabeled data following [48, 12]. For single-task teacher models, we adopt the same architecture and training schedules of Sparse R-CNN [38] and MaskFormer [5]. For object detection, we use a score threshold of 0.5 to select pseudo box labels. For segmentation, we use the score threshold of 0.3 to select segmentation masks. Pixels with lower scores are assigned with the ignore label. Thereafter, every image has annotations for all tasks and we train a multi-task student model with a weighted sum of all objectives for each task:

$$\mathcal{L}_{total} = \alpha_{det}\mathcal{L}_{det} + \alpha_{sem}\mathcal{L}_{sem} + \alpha_{driv}\mathcal{L}_{driv}, \tag{1}$$

where $\mathcal{L}_{det}, \mathcal{L}_{sem}, \mathcal{L}_{driv}$ are losses for object detection, semantic segmentation, and drivable area segmentation, respectively. [1] We include more details in Section 5.1.

**Zeroing loss** Some works [19, 46] simply zero losses for a particular task if the input image does not have the corresponding annotation.

**Batch-level Round-Robin** In this scheme [52, 22], each batch consists of samples that have the same type of annotations for a task and different tasks are repeated in a fixed order during training.

**Task Scheduling** This is a stochastic schedule that the task for each SGD update step is sampled from a distribution [22]. We mainly consider a uniform task sampling schedule that sampled from a uniform distribution, and a weighted task sampling schedule where the sampling weight for a task is proportional to the number of labeled images and training epochs for this task.

**Performance Comparison** As can be seen in Table 1 and Table 3, these methods surpass single-task teacher model by a clear margin, which indicates that the model benefits from multi-task learning. Among these baselines, self-training achieves satisfactory performance while being conceptually simple, thus we adopt it for the rest experiments in this paper.

---

[1]We conduct grid search in the range of [0.1, 1.0] with a step size of 0.1 to find the optimal loss weights setting and $\alpha_{det}, \alpha_{sem}, \alpha_{driv}$ are set to $1.0, 0.7, 0.7$, respectively. We fix the loss weights throughout the paper.

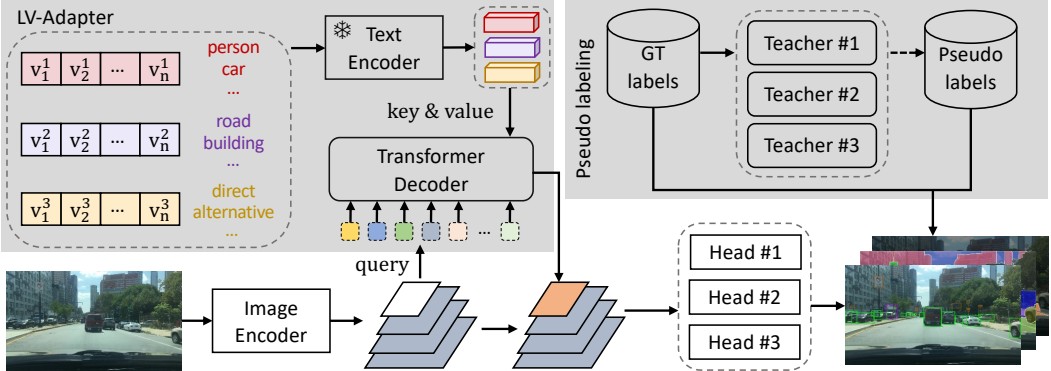

Figure 3: An overview of our proposed model. We first train three specialized teacher on labeled data to generate pseudo labels for each task. The multi-task model is then trained on both ground-truth and pseudo labels under the language-guided *pretrain-adapt-finetune* paradigm.

### 3.3 Pretrain-finetune for Multi-Task Learning

In this section, we adopt the identical training protocol and architecture to extensively investigate the performances of different types of pre-training methods, including task-oriented methods and vision-language pre-training, when transferred to the multi-task learning scenario. We revisit the technical details of these different schemes in the Appendix. We report the performance of the aforementioned methods in Table 2, and come to the following observations:

- Only SimCLR achieves decent performance on all three tasks and many methods encounter substantial degradation on pixel-level segmentation tasks. For semantic segmentation, MoCo-v1/v2 and DenseCL achieve the worst mIoU (degrade by 66%-83% v.s. ImageNet-pretrained). For drivable area segmentation, MoCo-v1/v2, DenseCL, and CLIP have the worst performance (degrade by nearly 12% v.s. ImageNet-pretrained).

- The pre-training paradigm does not seem to have an explicit correlation with the downstream performance, and even the task-oriented design in pre-training does not guarantee the transfer performance on the corresponding task type. For example, the segmentation-oriented DenseCL achieves sub-optimal performance on two segmentation tasks.

## 4 Effective Adaptation for Multi-Task Learning

### 4.1 Pretrain, Adapt, then Finetune

We attribute the degraded transferring performance of the state-of-the-art pre-training methods to two critical factors, i.e., optimization gap and architectural gap. Firstly, the high specialization of the pre-training methods and the heterogeneity of downstream tasks result in the optimization gap in the classic *pretrain-finetune* paradigm. To be specific, the objective for pre-training is usually a type of contrastive loss (e.g., InfoNCE [42]) while the fine-tuning stage is supervised by a weighted sum of several task-specific losses. Secondly, we notice that many models in Table 2 are pretrained with a convolutional head, e.g., Fast R-CNN detector [13] or convolution-based projection head. In contrast, we adopt the transformer-based MaskFormer head for segmentation tasks. The distinctions between convolution and transformer lie in the inductive bias (local v.s. global) and the internal representation structure [30] and we conjecture this plays a non-negligible role in transferring. We further experimentally verify this point in Appendix.

The limited transfer performance presented in Table 2 indicates that the 'universal' pre-training remains unsolved. Instead of redesigning another pre-training scheme that brings vast computation overhead, we aim to efficiently reuse the intact knowledge of these off-the-shelf pre-trained models with minimal modifications to the architecture. We draw inspiration from the recent progress of prompt-based learning, where prompt tuning emerges as a new alternative to fine-tuning. For example, P-Tuning v2 [25] matches the fine-tuning performance by only tuning learnable prompts

while freezing the large-scale language model. Following this philosophy, we propose a simple yet effective *pretrain-adapt-finetune* paradigm to mitigate the gap between the pre-training and fine-tuning stage. During the *adaptation* stage, we are given the pretrained model weights (e.g., ResNet-50) inherited from the *pre-training* stage. We aim to transform the model weights via a small amount of learnable parameters to adapt the knowledge of the pretrained weights towards multi-task scenarios. To this end, we freeze the parameters of the random initialized task-specific heads and the backbone, and tune the parameters of Feature Pyramid Network (FPN) supervised by the multi-task loss function in Equation 1. During *fine-tuning*, all parameters are activated and updated via gradient descent. We compare different schemes for multi-task learning in Figure 2. This *adaptation* stage characterizes several critical design choices: (1) This stage bridges the gap between *pre-training* and *fine-tuning* by including the pretrained weights and multi-task objectives simultaneously, and the frozen backbone prevents the pretrained weights from being spoiled before the *fine-tuning* stage. (2) Compared to the single-layer prompt/adapter in P-Tuning [20] and CLIP-Adapter [10], the FPN acts as a multi-scale adapter that is endowed with greater capacity and provides semantically stronger features for fine-grained downstream tasks. (3) Taking FPN as the adapter means that the model architecture for the *pretrain-finetune* and *pretrain-adapt-finetune* paradigm is identical and effective adaptation can be achieved without any modification to the architecture. We have experimented with more sophisticated adapters (e.g., scale-aware adapters, lightweight transformers) and observed marginally better results. Hence, we opt for the original architecture above for simplicity. We experimentally verify that our *pretrain-adapt-finetune* paradigm significantly improves the performance and stability of different kinds of pre-training methods (c.f. Section 5.2) without increasing the total training costs (c.f. Section 5.3).

### 4.2 Language-to-Vision Adapter

From the perspective of transfer learning, CLIP can serve as a complementary scheme to enhance the *pretrain-adapt-finetune* paradigm, since it can comprehend the concepts in natural language. Basically, the CLIP model excels in aligning the visual and language embeddings and some works [54, 31] claim that the textual features generated by CLIP have meaningful correspondence to the semantic regions in an image. We note that the result of CLIP in Table 2 only reuses the weights of its image encoder and discards the text encoder. Therefore, we take a step further to explicitly exploit the knowledge in the full CLIP model. We pursue to underpin the compatibility between the semantic concepts of each task and the image features and generate semantically stronger contexts for downstream tasks. The resulting model, named LV-Adapter, is outlined in Figure 3, and we elaborate on each ingredient in the followings.

The CLIP model adopts ResNet [17] and BERT [7] as the encoder for image and text, respectively. Formally, we denote the last output feature maps from ResNet stages as $\{\mathbf{x}_i\}_{i=2}^5$. Attentional pooling and $L_2$ normalization are applied to $\mathbf{x}_5 \in \mathbb{R}^{H_5 W_5 \times C_5}$ to produce the global image feature $\hat{\mathbf{I}}_e \in \mathbb{R}^{1 \times C_5}$ for zero-shot inference, which can be formulated as follows:

$$[\hat{\mathbf{I}}_e, \hat{\mathbf{x}_5}] = \text{L2\_NORM}(\text{MHSA}(\text{GAP}(\mathbf{x}_5) \oplus \mathbf{x}_5)), \tag{2}$$

where $\text{GAP}(\cdot)$ denotes global average pooling, $\text{MHSA}(\cdot)$ denotes multi-head self-attention [43], and $\oplus$ denotes concatenation operation. To extract textual features of each class, class-specific prompts are constructed via a generator function and are fed into the text encoder. We denote the normalized output features for $N$ classes as $\hat{\mathbf{T}}_e \in \mathbb{R}^{N \times C}$, which can be obtained as follow:

$$\hat{\mathbf{T}}_e = \text{L2\_NORM}(\text{TE}(\text{Gen}(\{\mathbf{n}_i\}_{i=1}^N))), \tag{3}$$

where $\text{Gen}(\cdot)$ is the generator function of class-specific prompts, TE is the text encoder, and $\{\mathbf{n}_i\}_{i=1}^N$ is the class name embeddings. Though $\hat{\mathbf{T}}_e$ and $\hat{\mathbf{I}}_e$ are well aligned in the image-text contrastive pre-training, they do not preserve any spatial details and are not readily applicable for downstream vision tasks. In contrast, the multi-level features output by FPN are semantically rich in the spatial dimension but are not directly aligned in the pre-training stage. Therefore, we pursue to learn task-specific prompts and propose a Language-to-Vision adapter to enhance the pixel-class correspondence.

**Learning Task-Specific Prompts.** The original generator function of CLIP outputs hand-crafted prompts using the predefined template, e.g., "a photo of a [CLS].". However, the performance is highly sensitive to the form of the template [29, 55], which may be sub-optimal when transferring to heterogeneous downstream tasks. Hence, we follow CoOp [55] to use learnable textual contexts in

Table 3: Results of single-task baselines and multi-task models with ResNet-50 backbone. SS and DA means semantic segmentation and drivable area segmentation. - indicates inapplicable.

| Setting | Method | mIoU (SS) | mIoU (DA) | mAP | AP50 | AP75 |
|---|---|---|---|---|---|---|
| Full | MaskFormer [5] | 57.1 | - | - | - | - |
| | MaskFormer [5] | - | 83.9 | - | - | - |
| | Sparse R-CNN [38] | - | - | 29.4 | 55.8 | 26.4 |
| | Self-training [12] | 61.8 | 84.4 | 30.1 | 56.6 | 27.6 |
| | **LV-Adapter (Ours)** | **63.1** | **84.9** | **31.1** | **58.2** | **28.4** |
| Disjoint-balance | MaskFormer [5] | 57.1 | - | - | - | - |
| | MaskFormer [5] | - | 78.1 | - | - | - |
| | Sparse R-CNN [38] | - | - | 18.6 | 37.8 | 15.6 |
| | Self-training [12] | 59.4 | 80.3 | 22.4 | 44.1 | 19.6 |
| | **LV-Adapter (Ours)** | **61.8** | **80.6** | **24.6** | **47.4** | **21.9** |
| Disjoint-normal | MaskFormer [5] | 57.1 | - | - | - | - |
| | MaskFormer [5] | - | 82.0 | - | - | - |
| | Sparse R-CNN [38] | - | - | 20.9 | 41.9 | 17.8 |
| | Self-training [12] | 60.3 | 83.1 | 24.9 | 48.1 | 22.2 |
| | **LV-Adapter (Ours)** | **62.2** | **83.7** | **26.4** | **50.5** | **23.7** |

prompting for each task. In spite of the inconsistent output formats of the three tasks (4-D box v.s. dense output), they both need to determine the category of boxes or pixels, and we incorporate the class names of each task in prompting. The task-specific prompting can be formulated as follows:

$$\hat{\mathcal{T}}_{e,i} = \text{L2\_NORM}(\text{TE}([\mathbf{v}^t, \mathbf{n}_i])), \tag{4}$$

where $\mathbf{n}_i$ refers to the embeddings of the class name belonging to a specific task, and $\mathbf{v}^t$ is the task-specific learnable contexts and the superscript indicates the task type.

**Enhancing Pixel-Class Correspondence.** To align the textural features and the dense FPN features, we propose a Language-to-Vision Adapter to incorporate the language priors into the visual features. Formally, we denote the last feature map of $P_5$ as $\mathbf{z}_5 \in \mathbb{R}^{H_5 W_5 \times C}$, and we aim to learn an adapter function $\mathcal{A}_{L \to V}$ to generate language-aware contexts for downstream tasks. We utilize the cross-attention mechanism in Transformer decoder [43] for Language-to-Vision adaptation:

$$\mathcal{A}_{L \to V}(\hat{\mathcal{T}}_e, \mathbf{z}_5) = \text{TransDecoder}(q = \mathbf{z}_5, k = \hat{\mathcal{T}}_e, v = \hat{\mathcal{T}}_e), \tag{5}$$

where the $q, k, v$ stands for query, key, and value. A single linear fully connected layer is used to adjust the channel number of $\hat{\mathcal{T}}_e$ and we omit it in Equation 5 for simplicity. We denote the output of Equation 5 as $\tilde{\mathbf{z}}_5 \in \mathbb{R}^{H_5 W_5 \times C}$. After adaptation, we simply substitute $\mathbf{z}_5$ with $\tilde{\mathbf{z}}_5$ to inject linguistic knowledge into the high-level features of FPN and leave the task-specific head design unchanged. We follow the same paradigm described in Section 4.1 when incorporating LV-Adapter: only the adapter module (i.e., LV-Adapter and FPN) is tuned during the *adaptation* stage. Moreover, we freeze the weights of the CLIP text encoder throughout the paper to preserve the language priors. We verify the efficacy of our LV-Adapter and disentangle the effect of different modules ablation in Section 5.3.

## 5 Experiments

### 5.1 Experiment Settings

In BDD100K, 70k training images are labeled for both object detection and drivable area segmentation, and only 7k training images are labeled for semantic segmentation. These two sets are not disjoint and about 3k images have complete annotations for all three tasks. As is described in [12], data annotation is one of the biggest challenges of training a multi-task model. In real-world scenarios, it is unrealistic to obtain complete types of annotations for all input images, especially in the traffic scene where fine-grained annotations are required. In this paper, we are interested in the performance of multi-task learning with different quantities of annotations. Thus, we mainly consider three scenarios that correspond to three different levels of annotation scarcity, namely, Disjoint-normal, Disjoint-balance,

and Full setting. We mainly work under the Disjoint-normal setting where the annotations of each task are disjoint, and the number of labeled images for each task is in decreasing order as follows: drivable area segmentation (20k), object detection (10k), semantic segmentation (7k). We provide the details of these three settings in the Appendix.

**Implementation Details.** The training epoch is fixed as 36, syncBN [28] is on, the learning rate at the *fine-tuning* stage is $2.5 \times 10^{-5}$, and weight decay is $1 \times 10^{-4}$. The image scale is $1280 \times$ (720, 600). No other data augmentation is used. We adopt the AdamW [27] optimizer. The warmup iteration is 1000 and the warmup factor is 0.01. For LV-Adapter, the learnable prompts are prepended to the class, and the length of prompts is 16. During the *adaptation* stage, the learning rate is set to $2.5 \times 10^{-4}$. The layer of transformer decoder in Equation 5 is 3. All experiments are conducted on servers with 8 Nvidia V100 GPU (32GB) cards and Intel Xeon Platinum 8168 CPU (2.70GHz).

## 5.2 Main Results

We present the results of our proposed paradigm in Table 2. The adapt and finetune epochs of the proposed paradigm in Table 2 are set as 6 and 30 respectively. As can be seen, with the conventional *pretrain-finetune* paradigm, the performances of the state-of-the-art self-supervised methods are unstable, especially in semantic segmentation and drivable area segmentation. For example, MoCo-v2 only achieves 10.3 mIoU on semantic segmentation. In contrast, under the *pretrain-adapt-finetune paradigm*, MoCo-v2 improves by a large margin in semantic segmentation (+50.9 in mIoU). Notably, MoCo-v1/v2 and DenseCL obtain an absolute improvement by over 40% mIoU in semantic segmentation. The results demonstrate that the *pretrain-adapt-finetune* paradigm significantly improves the overall performance across different types of pretrained models. We also implement experiments with popular convolutional heads, i.e., Faster R-CNN [32] and Semantic FCN [18], as in Appendix D.2. Results show that architecture is a core reason for the degradation. Moreover, we further showcase the efficiency of this paradigm in Section 5.3. We compare the results of our LV-Adapter to single-task baselines and the self-training baseline using the ImageNet pretrained model under three settings in Table 3. It demonstrates that LV-Adapter performs better on all metrics. Therefore, we conclude that the language-guided *pretrain-adapt-finetune* paradigm can effectively reduce the gap between the *pre-training* and *fine-tuning* stage by incorporating the linguistic knowledge to visual features.

## 5.3 Ablation Study

**Module design.** We compare the performance of different module designs in Table 4 using the pre-trained weights from CLIP. Row #1 refers to the performance of self-training with the *pretrain-finetune* paradigm using the pretrained weights of the CLIP image encoder. Row #2-#5 refer to the performance of the *pretrain-adapt-finetune* paradigm and row #3-#5 reuse the pretrained weights of CLIP text encoder. The adapt and finetune epochs for the pretrain-adapt-finetune paradigm are fixed as 1 and 5. The Prompt column refers to naive prompting without alignment between visual and textual features [31]: we compute class-specific textual features by prompting the CLIP text encoder (Equation 4) and compute the class activation maps by computing dot product between the textual features and $z_5$ after normalization, and the output maps are concatenated to $z_5$ and a $1 \times 1$ convolution layer is used to reduce the number of channel. The V2L column refers to the post-model module in [31], where the image context is used to prompt the language model and language-compatible class activation maps are concatenated back to $z_5$. In comparison, our Language-to-Vision Adapter (L2V column) directly incorporates linguistic knowledge into visual features and does not rely on dense activation maps and is thus more lightweight. As can be seen, when retrieving knowledge from pretrained text encoder via continuous prompt, the model improves in semantic segmentation by 0.3 mIoU (#2 vs #3). When we adopt the Vision-to-Language Adapter to update the text features, the performance degrades slightly in semantic segmentation (#3 vs #4). In contrast, our proposed LV-Adapter can better extract prior knowledge from CLIP and achieve the best results (+0.4 and +0.2 in mIoU in semantic segmentation and drivable area segmentation respectively).

**Number of epochs of the *adapt* and *finetune* stage.** We compare different configurations of *adapt* and *finetune* epochs in Table 5. When adapting with fewer epochs and fixing full epochs as 36, the performance improves gradually and the best configuration is 1 epoch for *adapt* and 35 epochs for *finetune*. Since we freeze the backbone and randomly initialized heads during the *adapt* stage, the number of trainable parameters is moderate. We can also see that with fixed finetune epochs, more

Table 4: Ablation study of the components of our proposed LV-Adapter.

| # | Prompt | V2L | L2V | mIoU (SS) | mIoU (DA) |
|---|--------|-----|-----|-----------|-----------|
| 1 | ✗ | ✗ | ✗ | 54.5 | 74.1 |
| 2 | ✗ | ✗ | ✗ | 61.5 | 83.5 |
| 3 | ✓ | ✗ | ✗ | 61.8 | 83.5 |
| 4 | ✓ | ✓ | ✗ | 61.3 | 83.6 |
| 5 | ✓ | ✗ | ✓ | **62.2** | **83.7** |

Table 5: Comparison of different configurations of *adapt* and *finetune* epochs.

| Adapt | Finetune | mIoU (SS) | mIoU (DA) | mAP |
|-------|----------|-----------|-----------|-----|
| 12 | 24 | 59.5 | 82.9 | 25.8 |
| 6 | 30 | 61.0 | 83.4 | 26.3 |
| 1 | 35 | 61.5 | 83.5 | 26.4 |
| 6 | 35 | 61.3 | 83.5 | 26.5 |
| 12 | 35 | 61.9 | 83.5 | 26.6 |

adapt epochs have little impact on the results. We hypothesize that fewer adapt stage is enough for the training paradigm since there are few training parameters (only the adapter) during the adapt stage. For a fair comparison with the baseline whose full epochs are 36, we fix the adapt and finetune epochs as 1 and 35 respectively.

# 6 Conclusion

In this paper, we present the first effort on revealing the degradation of state-of-the-art self-supervised models under the multi-task learning scenario in autonomous driving. To reduce the gap between the *pre-training* and *fine-tuning* stage, we propose a simple yet highly efficient *pretrain-adapt-finetune* paradigm, which boosts the performances of different self-supervised pretrained models by a large margin without increasing the overall training overhead. We further utilize the vision-language pre-training model CLIP as a complement to our proposed paradigm and propose LV-Adapter, which incorporates linguistic knowledge into visual features via learning task-specific prompts in a novel fashion. Extensive experiments showcase that (1) the *adapt* stage is the key to mitigate the deficiency of the dominating *pretrain-finetune* paradigm in multi-task learning, and (2) the language priors encoded by CLIP are of general benefits for multiple downstream tasks.

## Acknowledgements

We gratefully acknowledge the support of MindSpore[2], CANN (Computer Architecture for Neural Networks) and Ascend AI Processor used for this research.

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
