# Appendix for Effective Adaptation in Multi-Task Co-Training for Unified Autonomous Driving

## A    Experiment Setting

We introduce the details of three settings in our experiments.

**Disjoint-normal setting.** Under the same annotation effort or budget, the quantity of labels decreases with the increase of the complexity of labeling a task. Hence, in this setting, we consider a realistic scenario where each input image is labeled to only one task, namely, the annotations of each task are disjoint, and the number of labeled images for each task is in decreasing order as follows: drivable area segmentation (20k), object detection (10k), semantic segmentation (7k).

**Disjoint-balance setting.** We further consider a more difficult setting with the lowest quantity of annotations that corresponds to the scenarios of scarce annotations. There are 21k images in this setting and each task has 7k labeled images that are not overlapped with other tasks.

**Full-setting.** Full-setting refers to experimenting on all available annotations on $\sim$74k images in BDD100K and can be used to analyze the upper bound of different methods.

## B    Revisiting Different Types of Pre-training Methods

**Supervised pre-training** We select the weights pretrained on the ImageNet, which is the most widely adopted pretrained model in the past decade.

**Classification-oriented methods** Currently, the state-of-the-art classification-oriented pre-training methods are mainly based on contrastive learning and online clustering. During the pre-training phase, they produce image-level prediction using global features and minimize the distance between positive pairs while push the representations of negative pairs apart. This eliminates the need for the computation-intensive generation step in previous generative methods [3, 4]. However, they may lack spatial sensitivity for fine-grained tasks since the spatial details are not considered in their formulation. We select MoCo-v1 [10], MoCo-v2 [10], SimCLR [2], SwAV [1], BYOL [8] for comparison.

**Detection-oriented methods** DetCo [19] is specially designed for object detection and enforce contrastive learning between the global image and local patches with multi-level supervision. It achieves well performance on object detection while maintaining competitive classification accuracy.

**Segmentation-oriented methods** Different from the aforementioned ones, this type of method pursues pixel-level self-supervised learning for learning dense feature representations. For example, PixPro [20] propose a pixel-to-propagation consistency task, where two asymmetric pipelines are utilized to obtain positive pixel pairs. We select DenseCL [18] for comparison. We download self-supervised pretrained models from MMSelfSup repository[1].

## C    Discussion

We mainly discuss several closely related works on multi-task learning and highlight the differences in between in this section.

**Multi-Task Self-Training**. [7] focuses on the pre-training stage and claims that pre-training on a multi-task pseudo labeled dataset outperforms specialized supervised models and self-supervised models. Some of its findings back up our experiment results in Section 3. In [7], this pre-training paradigm is tailored for downstream **single-task learning** but not multi-task learning. Moreover, MuST conducts cross dataset pretraining on nearly 304M images with an enormous amount of pseudo labels, which incurs a huge computation overhead. In contrast, we show that the performance of a wide range of off-the-shelf pretrained models can be steadily improved without redesigning the resource-intensive pre-training stage or increasing the total training cost. We leave it for future work to explore the influence of this kind of pre-training paradigm on HMPL.

**Multi-Task Models.** We notice that several concurrent works [12, 14, 21, 13] have proposed multi-task models that cover a wide range of tasks. Our models differ notably mainly in two aspects. Firstly,

---

[1] https://github.com/open-mmlab/mmselfsup

Table 1: Comparison with handcrafted prompts under the disjoint-normal setting.

| Method | mAP | AP50 | AP75 | mIoU (SS) | mIoU (DA) |
|---|---|---|---|---|---|
| Prompt engineering | 26.1 | 50.1 | 23.4 | 61.4 | 83.3 |
| Prompt ensembling | 26.3 | 50.4 | 23.5 | 61.7 | 83.3 |
| Learned prompts | **26.4** | **50.5** | **23.7** | **62.2** | **83.7** |

Table 2: Comparisons of self-supervised models under the Disjoint-normal setting with ResNet-50 backbone and convolutional heads.

| Method | mAP | AP50 | AP75 | mIoU (SS) | mIoU (DA) |
|---|---|---|---|---|---|
| Supervised | 23.7 | 46.3 | 20.8 | 58.9 | 83.1 |
| MoCo-v1 | 24.2 | 46.9 | 21.5 | 59.6 | 83.1 |
| MoCo-v2 | 24.5 | 47.3 | 21.9 | 59.4 | 83.4 |
| SimCLR | 24.0 | 46.8 | 21.3 | 58.3 | 83.2 |
| SwAV | 24.0 | 47.5 | 20.7 | 59.7 | 82.8 |
| BYOL | 23.6 | 46.3 | 20.3 | 59.8 | 83.0 |
| DetCo | 24.1 | 46.8 | 21.5 | 59.6 | 83.3 |
| DenseCL | 24.7 | 47.7 | 22.0 | 59.8 | 83.4 |
| PixPro | 24.2 | 46.9 | 21.6 | 60.2 | 83.5 |

we perform *heterogeneous* multi-task learning where each task has different objectives and evaluation metrics, which increases the optimization difficulty. Secondly, we propose general approaches to improve the overall multi-task performance, while other works propose task-specific or model-specific methods instead.

**Task Grouping.** Task grouping approaches [17, 5] are mainly concerned with determining task groupings to avoid negative transfer between tasks. In this paper, we hypothesize that in the scenario of autonomous driving, the perception tasks are implicitly correlated with each other. From this perspective, task grouping approaches are complementary to ours since once the grouping is decided, the tasks within a group are trained together in a multi-head architecture and our proposed module can be seamlessly integrated.

# D   More Ablation Studies

## D.1   Comparison with Handcrafted Prompts

We compare learned prompts with traditionally handcrafted prompts as in Table 1. CLIP [15] designs several handcrafted prompts for zero-shot classification, such as "a photo of [CLASS].". However, these prompts are not suitable for object detection and semantic segmentation. Inspired by [9], we adopt a better prompt "there is a [CLASS] in the scene." for prompt engineering. We follow [15] to ensemble multiple prompt templates, which is also a popular method for zero-shot classification. As shown in Table 1, learned prompts perform better than prompt engineering and prompt ensembling. We conjecture that learned prompts encode more task-related meanings beyond the handcrafted templates.

## D.2   Influence of Head Architecture

To verify the influence of head architectures in multi-task learning, we build a new multi-task model based on Faster R-CNN [16] and Semantic FCN [11], which retains only convolution operators in heads. We report the performance of different pre-training methods in Table 3. As can be seen, their performance becomes much more stable, but is inferior to the transformer-based architecture in Table 2 in paper. The results indicate that the architectural gap is one of the core reasons for the degradation of these models.

To verify the effectiveness of our proposed adapt stage across different architectures, we provide results of the pretrain-adapt-finetune paradigm for popular convolutional heads (Faster R-CNN and Semantic FPN) in Table 3:

Table 3: Comparisons of different training schemes for self-supervised models with convolutional heads.

| Method | Scheme | mIou (SS) | mIoU (DA) | mAP |
|--------|--------|-----------|-----------|-----|
| SimCLR | pretrain-finetune | **58.3** | 83.2 | 24.0 |
|        | pretrain-adapt-finetune | **58.3** | **83.3** | **24.6** |
| DetCo  | pretrain-finetune | 59.6 | 83.3 | 24.1 |
|        | pretrain-adapt-finetune | **59.7** | **83.4** | **24.8** |

From the results, we conclude that the pretrain-adapt-finetune paradigm can boost performance on all tasks consistently. However, the improvement is limited, since there is less discrepancy between these CNN-based heads and pretrained models.

## D.3  Influence of Adapter Design

We also carry out additional experiments on different adaptation module designs inspired by [6]. Five more different designs are considered, including 1) pre-adapter: inserting two convolutional layers before the FPN; 2) post-adapter: adding two convolutional layers after the FPN; 3) pre-adapter (learnable scalar): pre-adapter with residual connection by summing the input and output feature weighted by the learnable scalar which is initialized with a small number 0.001; 4) pre-adapter (learnable vector): pre-adapter with residual connection by summing the input and output feature weighted by the learnable vector which is initialized with 0.001. Note that all experiments are trained with the same pseudo labels under the disjoint-normal setting and use the SimCLR pretrained model to initialize the backbone.

Table 4: Comparison of different adapter designs.

| Adapter | mIoU (SS) | mIoU (DA) | mAP |
|---------|-----------|-----------|-----|
| FPN | 60.1 | 83.5 | 25.2 |
| pre-adapter | 60.2 | 83.4 | 25.4 |
| post-adapter | 60.2 | 83.5 | 25.4 |
| pre-adapter (learnable scalar) | 59.9 | 83.6 | 25.5 |
| pre-adapter (learnable vector) | 59.8 | 83.6 | 25.1 |

From the results, we can observe that different adapters achieve comparable results. For simplicity, we choose the simple FPN adapter as the baseline.

## D.4  Language Prior for Single Task

We further conduct the experiments to validate our LV-Adapter equipped the language prior on single-task models under the disjoint-balance setting. Results are shown in Table 5.

Table 5: Comparisons on single-task setting.

| Model | mIoU (SS) | mIoU (DA) | mAP |
|-------|-----------|-----------|-----|
| MaskFormer | 57.1 | - | - |
| + LV-Adapter | **61.3**$^{+4.2\%}$ | - | - |
| MaskFormer | - | 78.1 | - |
| + LV-Adapter | - | **81.6**$^{+3.5\%}$ | - |
| Sparse R-CNN | - | - | 18.6 |
| + LV-Adapter | - | - | **22.4**$^{+3.8\%}$ |

We can see that equipped with language prior, LV-Adapter improves single-task models by a large margin (+4.2% mIoU in semantic segmentation, +3.5% mIoU in drivable area segmentation, and +3.8% mAP in object detection), indicating that the language knowledge can also serve as a strong prior to enhance the visual representation.

## E   Experiments on NuImages Dataset

We further conduct experiments on NuImages dataset[2], which contains full labels on both object detection and semantic segmentation. Results are shown in the following table 6. Note that the multi-task baseline is trained with FPN as the adapter under the pretrain-finetune paradigm.

Table 6: Comparisons of single-task and multi-task models on NuImages dataset.

| Method | mAP | AP50 | AP75 | mIoU |
|---|---|---|---|---|
| Sparse R-CNN | 46.6 | 72.8 | 49.4 | - |
| MaskFormer | - | - | - | 55.8 |
| Multi-task | 47.1 | 73.5 | 49.9 | 53.3 |
| LV-Adapter | **50.3** | **76.8** | **54.3** | **56.0** |

As shown in the table, the multi-task model performs worse than single-task models in semantic segmentation. Our proposed LV-Adapter achieves the best on all tasks, gaining 3.2% mAP and 2.7% mIoU improvement in object detection and semantic segmentation respectively compared with the multi-task baseline. This verifies the effectiveness of our method.

## F   Potential Negative Social Impact

Our method has no ethical risk on dataset usage and privacy violation since all benchmarks are public.

## G   Limitations

Due to the limited amount and the scarcity of annotations, we only study three tasks in parallel. A more diversified task set with more annotations will further benefit the learning of multi-headed architecture. Besides, since the MuST model [7] is not released and it is impractical for us to conduct resource-intensive multi-task pretraining with limited resources, the influence of multi-task pretraining on multi-task transfer learning is yet to be explored and is out of the scope of this paper.