# OpenReview forum: "Effective Adaptation in Multi-Task Co-Training for Unified Autonomous Driving"
_NeurIPS.cc/2022/Conference — NeurIPS 2022 Accept_

### Official Review · Reviewer_5cHS · 2022-07-09

**Rating:** 6
**Confidence:** 4
**Soundness:** 3 good
**Presentation:** 3 good
**Contribution:** 3 good

**Summary:**

This paper proposes a pretrain-adapt-finetune paradigm to train a multi-task network from a well-pretrained network which improves the overall performance compared to the conventional pretrain-finetune paradigm. During the adaption, authors propose to learn a multi-scale adapter. To better use textual modeling in the recent CLIP model, authors further proposes a LV-adapter via prompting learning to incorporate the language priors. The paper experiments on 8 different pretrained models to show their proposed diagram learns a better MTL model from the given pretrained network.

**Questions:**

1. Could you elaborate more on the experiment settings of Table 4 and Table 5? How do the experiments in Table 4 and 5 relate to those in Table 2?

2. Have you tried with other datasets or other types of tasks? In this paper, experiments mainly focus on the segmentation and detection tasks. What the performance would be when we have other types of tasks, e.g. classification?

3. In Table 3, authors show LV-Adapter is better than self-training. In my understanding, self-training is somehow orthogonal to LV-adapter. Specifically, we can use self-training to generate pseudo labels and train LV-adapter with the available gt labels and pseudo-labels. In this case, what is performance of the LV-Adapter with self-training? Will LV-Adapter keep improving with high-quality pseudo labels?

**Limitations:**

They have included in the supplementary

**Strengths And Weaknesses:**

Strengths

1. The pretrain-adapt-finetune diagram is shown effective with 8 different SoTA pretrained networks with various pretraining goals.

2. It is novel to learn a multi-task model for vision tasks from CLIP via prompt learning.

3. The adaption stage is cheap based on the ablation studies.

Weaknesses:

1. The experiment settings for Table 4 and Table 5 are not clearly stated in the paper. For instance, which pretrained model is used in for Table 5? What tasks are considered in Table 4? Why are the best results of Table 4 and 5 not matching the results in Table 2?

2. Experiments are only limited to BDD100k dataset

3. There is no ablation studies related to the different designs of multi-scale adapter. Can they be replaced with adapters, such as residual adapters?

---

> ### Author Response · Authors · 2022-08-02
> **Responses to Reviewer 5cHS**
>
> We thank the reviewer for the great suggestion. We are glad to hear that you appreciate the effective pretrain-adapt-finetune paradigm, the novel prompt learning for the multi-task model, and the cheap adaptation stage. In the following, we will address your concerns:
>
> - **Experiments on other dataset or task:** "Experiments are only limited to BDD100k dataset. What the performance would be when we have other types of tasks, e.g. classification?"
> - **Ablation studies on different adapter designs:** "There is no ablation studies related to the different designs of multi-scale adapter."
> - **Settings of Table 4 and Table 5:** "Which pretrained model is used in for Table 5? What tasks are considered in Table 4? Why are the best results of Table 4 and 5 not matching the results in Table 2?"
> - **LV-Adapter vs. self-training:** "What is performance of the LV-Adapter with self-training? Will LV-Adapter keep improving with high-quality pseudo labels?"
>
> **1. Experiments on other dataset or task**
>
> As suggested, we further conduct experiments on NuImages dataset [1], which contains full labels on both object detection and semantic segmentation. Results are shown in the following table. Note that the multi-task baseline is trained with FPN as the adapter under the pretrain-finetune paradigm.
>
> | Method       | mAP  | AP50             | AP75      | mIoU |
> |--------------|------|------------------|-----------|------|
> | Sparse R-CNN | 46.6 | 72.8             | 49.4      | -    |
> | MaskFormer | -    | -                | -         | 55.8 |
> | Multi-task | 47.1 | 73.5 | 49.9 | 53.3 |
> | LV-Adapter | **50.3** | **76.8** | **54.3** | **56.0** |
>
> As shown in the table, the multi-task model performs worse than single-task models in semantic segmentation.
> Our proposed LV-Adapter achieves the best on all tasks, gaining 3.2% mAP and 2.7% mIoU improvement in object detection and semantic segmentation respectively compared with the multi-task baseline. This verifies the effectiveness of our method.
>
> Besides, this paper focuses on studying the multi-task problems in autonomous driving, in which tasks mainly are detection-based and segmentation-based. Therefore, we did not consider other tasks (e.g., classification) in this paper.
>
> **2. Ablation studies on different adapter designs**
>
> We clarify that apart from the FPN adapter, we already propose a new LV-Adapter in Section 4.2 of this paper, which explicitly exploits the knowledge in the CLIP model and enhances task-specific image features. Experiments show that our LV-Adapter performs better than FPN with the pretrain-adapt-finetune paradigm as in Table 4 in the paper (row 5 vs. row 2), with +0.7% mIoU in semantic segmentation and +0.2% mIoU in drivable area segmentation.
>
> As suggested, we also carry out additional experiments on different adaptation module designs inspired by [2].
> Five more different designs are considered, including 1) pre-adapter: inserting two convolutional layers before the FPN; 2) post-adapter: adding two convolutional layers after the FPN; 3) pre-adapter (learnable scalar): pre-adapter with residual connection by summing the input and output feature weighted by the learnable scalar which is initialized with a small number 0.001; 4) pre-adapter (learnable vector): pre-adapter with residual connection by summing the input and output feature weighted by the learnable vector which is initialized with 0.001.
> Note that all experiments are trained with the same pseudo labels under the disjoint-normal setting and use the SimCLR pretrained model to initialize the backbone.
>
> | Adapter | mIoU (SS)    | mIoU (DA) | mAP  |
> |---------| ------- |-----------|------|
> | FPN   | 60.1 | 83.5      | 25.2 |
> | pre-adapter   | 60.2 | 83.4      | 25.4 |
> | post-adapter   | 60.2 | 83.5      | 25.4 |
> | pre-adapter (learnable scalar)   | 59.9 | 83.6      | 25.5 |
> | pre-adapter (learnable vector)  | 59.8 | 83.6 | 25.1 |
>
> From the results, we can observe that different adapters achieve comparable results. For simplicity, we choose the simple FPN adapter as the baseline. With task-specific prompts to extract useful language prior, our LV-Adapter further improve performance by a large margin (62.2% mIoU in semantic segmentation, 83.7% mIoU in drivable area segmentation, and 26.4 mAP in object detection).

---

> > ### Author Response · Authors · 2022-08-02
> > **Responses to Reviewer 5cHS (Continued)**
> >
> > **3. Settings of Table 4 and Table 5**
> >
> > The detailed settings of Table 4 and Table 5, and the correspondence with Table 2 are listed as follows:
> >
> > - Table 4 and Table 5 are both conducted under the disjoint-normal setting, and adopt pretrained weights from CLIP to initialize the backbone. Therefore, the third row of Table 5 corresponds to the second row of Table 4 (w/ adapt stage).
> > - The first row of Table 4 (w/o adapt stage) corresponds to the penultimate row of Table 2.
> > - The adapt and finetune epochs in Table 2 are set as 6 and 30 respectively to ensure a fair comparison with other pre-trained models. Thus the last row of Table 2 corresponds to the second row of Table 5.
> >
> > There is a minor mistake (83.7->83.5) in the mIoU (DA) of the third row of Table 5 and we have fixed it in the revision.
> >
> > **4. LV-Adapter vs. self-training**
> >
> > Sorry for the confusion, we clarify LV-adapter in Table 3 uses the same pseudo labels as in self-training. LV-adapter and self-training are both trained with the available GT labels and pseudo-labels.
> > We have made this setting clear in the revision.
> >
> > As shown in Table 3 in the main paper, when equipping self-training with our proposed LV-Adapter, we can further obtain a significant performance improvement, i.e., +1.9% mIoU in semantic segmentation, +0.6% mIoU in drivable area segmentation, and +1.5% mAP in object detection.
> > Besides, self-training also achieves better results compared with all single-task models in Table 3 in paper. Thus the quality of pseudo labels is high enough. We can also use the trained LV-Adapter to generate pseudo labels of higher quality, but the cost is high. And self-training is not our focus in this work.
> >
> > [1] https://www.nuscenes.org/nuimages
> >
> > [2] Gao et al. CLIP-Adapter: Better Vision-Language Models with Feature Adapters. arXiv preprint arXiv:2110.04544, 2021.

---

### Official Review · Reviewer_kNyY · 2022-07-11

**Rating:** 5
**Confidence:** 3
**Soundness:** 3 good
**Presentation:** 3 good
**Contribution:** 2 fair

**Summary:**

With the success of self-supervised learning, this paper attempts to make better use of the pre-trained model under the multi-task learning scenario in autonomous driving.
This paper proposed a pretrain-adapt-finetune paradigm, which boosts and performance of different self-supervised pre-trained models.
It further utilizes the vision-language pre-training model CLIP and propose LV-Adapter. The experiments show that the language priors encoded by CLIP can have some benefits for multiple downstream tasks.



**Questions:**

1. It would be better to compare your methods with previous multi-task learning methods, maybe Table 1 should have your approach for comparison.
2. It would be better for Table 2 to have a supervised baseline. As you mention in Appendix D.2, architecture is one of the core reasons for the degradation. From the results, we can see the degradation usually happens in semantic seg, the reason is the transformer-based segmentation head. Therefore, the effectiveness of ‘adapt’ is not well reflected.
3. Table 5 shows that fewer adapt stage (1) can have the best performance. I wonder the results of longer adapt stage with fixed fine-tune stage.
4. If the annotations of different tasks are not aligned, I wonder the realization of co-training process. This means using the pre-trained model to make pseudo labels and then training together? Does the gain come from training together or from more data utilization?
5. Why the Language prior can help the model? Is it useful only under the condition of multi task learning, or can it also assist the learning of each sub task alone?


**Limitations:**

The author discuss them in the appendix.

**Strengths And Weaknesses:**

Strengths:
The paper is well-organized and the studied problems have practical value.
The author tried many pre-trained methods, the analysis and experiment are sufficient to verify the proposed viewpoint.

Weaknesses:
The novelty is moderate, both the pretrain-adapt-finetune paradigm and using language priors is not new in this field, but it is worthy to explore them under the multi-task learning scenario in autonomous driving.
There are some confusing points in the experiments, I listed them in the question part.

---

> ### Author Response · Authors · 2022-08-02
> **Responses to Reviewer kNyY**
>
> We thank the reviewer for detailed and constructive feedback. We are glad to hear that you appreciate the practical studied problems and our sufficient analysis and experiment. In the following, we will address all your questions:
>
> - **Results of our method under the disjoint-normal setting:** "Table 1 should have the proposed method for comparison with previous multi-task learning methods."
> - **Supervised baseline for Table 2:** "It would be better for Table 2 to have a supervised baseline."
> - **Effectiveness of paradigm:** "Architecture is one of the core reasons for the degradation, and the effectiveness of ‘adapt’ is not well reflected in Appendix D.2."
> - **Longer adapt stage with fixed finetune stage:** "The results of longer adapt stage with fixed fine-tune stage."
> - **The realization and performance of co-training process:** "Whether use the pre-trained model to make pseudo labels and then training together? Does the gain come from training together or from more data utilization?"
> - **The impact of language prior for multi tasks and single task:** "Why the language prior can help the model? Is it useful only under the condition of multi task learning, or can it also assist the learning of each sub task alone?"
>
> **1. Results of our method under the disjoint-normal setting**
>
> Thanks for your suggestion and we have added the result of our approach in Table 1 for comparison in revision.
> Specifically, compared with the previous best performance for every single task, our LV-Adapter gains 1.5% mIoU, 0.4% mIoU, and 1.5% mAP improvement in semantic segmentation, drivable area segmentation, and object detection respectively with the proposed pretrain-adapt-finetune paradigm and language prior.
>
> **2. Supervised baseline for Table 2**
>
> As suggested, we build the supervised baseline with the same architecture and initialize it with weights from ImageNet supervised pretrained model.
> Results of the supervised baseline are shown in the following table.
>
> | Scheme  | mIoU (SS)    | mIoU (DA) | mAP  |
> |---------| ------- |-----------|------|
> | pretrain-finetune     | 60.3 | 83.1      | 24.9 |
> | pretrain-adapt-finetune | 60.3 | 83.4 | 24.6 |
>
> The pretrain-adapt-finetune paradigm achieves comparable performance compared with the baseline, which may be caused by the pretraining pipeline (e.g., classification task). We hypothesize that the information learned from image classification is more compatible for our downstream tasks than other contrastive pretrained models.
>
> **3. Effectiveness of paradigm**
>
> Since most state-of-the-art methods for single tasks (e.g., detection and segmentation) are transformer-based, we choose transformer-based heads to build our model.
> To verify the effectiveness of our proposed adapt stage across different architectures, we provide results of the pretrain-adapt-finetune paradigm for popular convolutional heads (Faster R-CNN and Semantic FPN) in the following table:
>
> | Method | Scheme                  | mIoU (SS)      | mIoU (DA)      | mAP            |
> |---------|-------------------------|----------------|----------------|----------------|
> | SimCLR | pretrain-finetune  | **58.3**  | 83.2  | 24.0  |
> | SimCLR |  pretrain-adapt-finetune | **58.3**  |  **83.3** |  **24.6** |
> | DetCo | pretrain-finetune  | 59.6  | 83.3  | 24.1  |
> | DetCo |  pretrain-adapt-finetune |  **59.7** |  **83.4** |  **24.8** |
>
> From the results, we conclude that the pretrain-adapt-finetune paradigm can boost performance on all tasks consistently. However, the improvement is limited, since there is less discrepancy between these CNN-based heads and pretrained models.
>
> **4. Longer adapt stage with fixed finetune stage**
>
> We conduct experiments with longer adapt stage and fixed finetune stage under the disjoint-normal setting. Results are provided in the table:
>
> | Adapt | Finetune | mIoU (SS)      | mIoU (DA) | mAP  |
> |-------|----------|----------------|-----------|------|
> | 1     | 35       | 61.5 | 83.5      | 26.4 |
> | 6     | 35       | 61.3 | 83.5      | 26.5 |
> | 12    | 35       | 61.9 | 83.5 | 26.6 |
>
> From the table, we can see that with fixed finetune epochs, the different number of adapt epochs have little impact on the results. We hypothesize that fewer adapt stage is enough for the training paradigm since there are few training parameters (only the adapter) during the adapt stage.

---

> > ### Author Response · Authors · 2022-08-02
> > **Responses to Reviewer kNyY (Continued)**
> >
> > **5. The realization and performance of co-training process**
> >
> > The realization of the co-training process consists of 1) first training single-task models to generate pseudo labels for unlabeled data; 2) training both annotated labels and pseudo labels together.
> >
> > We summarize the experimental results in the main paper to quantificationally analyze the performance gain for three aspects: 1 zeroing loss, in which we train the multi-task model with only annotated data; 2 self-training, in which we train the same multi-task model with both annotated data and pseudo labels; 3
> > LV-Adapter, in which we equip the self-training setting with our proposed pretrain-adapt-finetune paradigm and language-aware adapter modules.
> > We summarize the results of single-task models, zeroing loss, self-training, and our LV-Adapter from Table 1 and Table 3 in the paper as follows:
> >
> > | Model            | mIoU (SS) | mIoU (DA) | mAP  |
> > |------------------|-----------|-----------|------|
> > | MaskFormer | 57.1      | -         | -    |
> > | MaskFormer | -         | 82.0      | -    |
> > | Sparse R-CNN | -         | -         | 20.9 |
> > | 1 Zeroing loss | 59.4 | 83.3      | 23.0 |
> > | 2 Self-training | 60.3 | 83.1      | 24.9 |
> > | 3 LV-Adapter | **62.2** | **83.7** | **26.4** |
> >
> > As shown in the Table, we can observe that performance gains come from:
> > single task -> model 1: +2.3% mIoU (SS), +1.3% mIoU (DA), and +2.1% mAP; model 1 -> model 2: +0.9% mIoU (SS), and +1.9% mAP; model 2 -> model 3: +1.9% mIoU (SS), +0.6% mIoU (DA), and +1.5% mAP.
> >
> > **6. The impact of language prior for multi tasks and single task**
> >
> > The language prior is useful can be attributed to the fusion of visual features and textual features.
> > Since the CLIP model excels in aligning the visual and language embeddings, the textual features generated by CLIP have meaningful correspondence to the semantic regions in an image [1,2]. Therefore, LV-Adapter can exploit the knowledge in the full CLIP model and then enhance the general visual representation by the language prior.
> >
> > Besides, the language prior can be also helpful to assist the learning of each sub-task alone.
> > We further conduct the experiments to validate our language prior on single-task models under the disjoint-balance setting. Results are shown in the following Table:
> >
> > | Model                     | mIoU (SS) | mIoU (DA) | mAP |
> > |---------------------------|-----------|-----------|----|
> > | MaskFormer                | 57.1      | -         | -  |
> > | MaskFormer + LV-Adapter   | **61.3**      | -         | -  |
> > | MaskFormer                | -         | 78.1      | -  |
> > | MaskFormer + LV-Adapter   | -         | **81.6**      | -  |
> > | Sparse R-CNN              | -         | -         | 20.9 |
> > | Sparse R-CNN + LV-Adapter | -         | -         | **22.4** |
> >
> > We can see that equipped with language prior, LV-Adapter improves single-task models by a large margin (+4.2% mIoU in semantic segmentation, +3.5% mIoU in drivable area segmentation, and +3.8% mAP in object detection), indicating that the language knowledge can also serve as a strong prior to enhance the visual representation.
> >
> > [1] Zhou et al. Denseclip: Extract free dense labels from clip. arXiv preprint arXiv:2112.01071, 2021.
> >
> > [2] Zhou et al. Learning to prompt for vision-language models. arXiv preprint arXiv:2109.01134, 2021.

---

> > > ### Comment · Reviewer_kNyY · 2022-08-09
> > > **Thank you for the response**
> > >
> > > Thank you for your subsequent experiments and analysis. I have no more questions, just some suggestions.
> > > Results in Answer 6 are quite convincing to see the effectiveness of language prior.
> > > Since the degradation mainly comes from the inconsistency of the architecture, it would be better to clarify in the paper.

---

> > > > ### Author Response · Authors · 2022-08-09
> > > > **Thank you for the suggestions**
> > > >
> > > > Thank you for your response and suggestions. We have added the experiments and analysis in Appendix, and clarified that architecture is a core reason for the degradation in L309 in the paper.

---

### Official Review · Reviewer_72BH · 2022-07-11

**Rating:** 6
**Confidence:** 2
**Soundness:** 3 good
**Presentation:** 3 good
**Contribution:** 3 good

**Summary:**

This paper is the first to study the use of self-supervised pretrained models in multi-task learning scenarios under autonomous driving settings. It points out that the conventional pretrian-finetune scheme is not effective, and instead, it proposes the pretrain-adapt-finetune paradigm. The adaption is conducted on the FPN, and language-to-vision adaptation with pretrained language model further improve the performance.

**Questions:**

Please see the Weaknesses section.

**Limitations:**

No additional limitations or potential negative societal impact.


**Strengths And Weaknesses:**

Strengths:
1. Multi-task training and inference is a critical research direction for the autonomous driving industry due to its multi-task nature of daily operations. I am pleased to see a pioneering effort on this topic.
2. The proposed pretrain-adapt-finetune scheme is effective. The discussion on using FPN as the adaption module is especially convincing.
3. The inclusion of a language model in assisting learning of downstream tasks is interesting.

Weaknesses:
1. Line 173-174: it is stated “these methods surpass single-task teacher model by a clear margin”, which rows are for single-task teacher models?
2. Although FPN seems to be a good choice for adaptation, it may be more convincing to try other parts and combinations involving the backbone and the head.
3. It is interesting to see that L2V works, and it is helpful to discuss more in depth that why L2V works better than V2L, probably from the angle that what role the language model plays.
4. On learning task-specific prompts, what are the typical learned prompts for each of the tasks?

---

> ### Author Response · Authors · 2022-08-02
> **Responses to Reviewer 72BH**
>
> We thank the reviewer for the insightful comment. We are glad to hear that you appreciate our proposed pretrain-adapt-finetune scheme and the inclusion of the language model in assisting the learning of downstream tasks. In the following, we will address all your questions:
>
> - **The advantage of L2V module:** "Discuss more in depth that why L2V works better than V2L."
> - **Different adaptation module design:** "Try other parts and combinations involving the backbone and the head."
> - **What are task-specific prompts:** "On learning task-specific prompts, what are the typical learned prompts for each of the tasks?"
> - **Comparison with single-task teacher models:** "Which rows of results are for single-task teacher models?"
>
> **1. The advantage of L2V module**
>
> The reason that L2V works better can be attributed to the fusion of visual features and textual features.
> Since the CLIP model excels in aligning the visual and language embeddings, the textual features generated by CLIP have meaningful correspondence to the semantic regions in an image [1,2]. Therefore, L2V module can further exploit the knowledge in the full CLIP model and then enhance the general visual representation by the language prior.
>
> On the other hand, in V2L module, the image context is used to prompt the language model and produce language-compatible dense class activation maps as priors.
> However, the generated class activation map can be inaccurate since the CLIP is trained with image-text supervision, and is hard to deal with the pixel-wise prediction tasks, especially for complex autonomous driving scenarios.
>
> As shown in Table 4 in the paper, L2V performs better than V2L and improves semantic segmentation and drivable area segmentation by 0.9% and 0.1% mIoU respectively.
>
> **2. Different adaptation module designs**
>
> We clarify that apart from the FPN adapter, we already propose a new LV-Adapter in Section 4.2 of this paper, which explicitly exploits the knowledge in the CLIP model and enhances task-specific image features. Experiments show that our LV-Adapter performs better than FPN with pretrain-adapt-finetune paradigm as in Table 4 in the paper (row 5 vs. row 2), with +0.7% mIoU in semantic segmentation and +0.2% mIoU in drivable area segmentation.
>
> As suggested, we also carry out additional experiments on different adaptation module designs inspired by [3].
> Five more different designs are considered, including 1) pre-adapter: inserting two convolutional layers before the FPN; 2) post-adapter: adding two convolutional layers after the FPN; 3) pre-adapter (learnable scalar): pre-adapter with residual connection by summing the input and output feature weighted by the learnable scalar which is initialized with a small number 0.001; 4) pre-adapter (learnable vector): pre-adapter with residual connection by summing the input and output feature weighted by the learnable vector which is initialized with 0.001.
> Note that all experiments are trained with the same pseudo labels under the disjoint-normal setting and use the SimCLR pretrained model to initialize the backbone.
>
> | Adapter | mIoU (SS)    | mIoU (DA) | mAP  |
> |---------| ------- |-----------|------|
> | FPN   | 60.1 | 83.5      | 25.2 |
> | pre-adapter   | 60.2 | 83.4      | 25.4 |
> | post-adapter   | 60.2 | 83.5      | 25.4 |
> | pre-adapter (learnable scalar)   | 59.9 | 83.6      | 25.5 |
> | pre-adapter (learnable vector)  | 59.8 | 83.6 | 25.1 |
>
>
> From the results, we can observe that different adapters achieve comparable results. For simplicity, we choose the simple FPN adapter as the baseline. With task-specific prompts to extract useful language prior, our LV-Adapter further improve performance by a large margin (62.2% mIoU in semantic segmentation, 83.7% mIoU in drivable area segmentation, and 26.4 mAP in object detection).
>
> **3. What are task-specific prompts**
>
> Prompt-based transfer learning has shown promising performances for a large number of downstream tasks [2,4]. Inspired by [2], we leverage continuous soft prompt to extract task-specific contexts from the pretrained model.
> Specifically, task-specific prompts are learnable vectors for different tasks. The detailed definition is included in the Section 4.2 in the paper.
> We totally have three prompts for object detection, semantic segmentation, and drivable area segmentation respectively. These prompts should be dissimilar due to the difference among tasks. We compute cosine similarities (range from -1 ~ 1) between different task prompts, and show the result as follows: cosine similarity for object detection and semantic segmentation, object detection and drivable area segmentation, semantic segmentation and drivable area segmentation are -0.69, -0.18, and -0.27 respectively. From the results, we can observe that task-specific prompts are quite different for each task.
> The learned prompts further reuse the language knowledge from the pretrained model to enhance the task-specific visual feature.

---

> > ### Author Response · Authors · 2022-08-02
> > **Responses to Reviewer 72BH  (Continued)**
> >
> > **4. Comparison with single-task teacher models**
> >
> > The performance of single-task teacher models is shown in row 11-13 of Table 3 in the main paper since Table 1 is conducted under the disjoint-normal setting. Specifically, single-task teacher models obtain 57.1% mIoU in semantic segmentation, 82.0% in drivable area segmentation, and 20.9% mAP in object detection. We can observe that multi-task models surpass single-task models by a large margin.
> >
> > [1] Zhou et al. Denseclip: Extract free dense labels from clip. arXiv preprint arXiv:2112.01071, 2021.
> >
> > [2] Zhou et al. Learning to prompt for vision-language models. arXiv preprint arXiv:2109.01134, 2021.
> >
> > [3] Gao et al. CLIP-Adapter: Better Vision-Language Models with Feature Adapters. arXiv preprint arXiv:2110.04544, 2021.
> >
> > [4] Yao et al. Cpt: Colorful prompt tuning for pre-trained vision-language models. arXiv preprint arXiv:2109.11797, 2021.

---

### Author Response · Authors · 2022-08-02
**Summary of responses - thanks to all reviewers for thorough and insightful feedback**

We thank all reviewers for their time, insightful suggestions, and valuable comments. We are glad that reviewers find our work effective (Reviewer 72BH & 5cHS) and appreciate our sufficient experiments and analyses (Reviewer kNyY). Following the constructive suggestions and comments of the reviewers, we have revised our manuscript and provided more experimental results to further analyze the proposed method. The main changes we made include:

- In Appendix D.3, we add an ablation study comparing different adapter designs.
- In Appendix D.2, we add an ablation study on convolutional heads with the pretrain-adapt-finetune paradigm.
- In Appendix D.4, we add an ablation study on the performance of LV-Adapter against single-task models under the single-task setting.
- In Appendix E, we add experiments on NuImages dataset comparing our LV-Adapter with single-task and multi-task baselines.
- In the paper revision, we add the result of our LV-Adapter to Table 1.
- In the paper revision, we add results of the proposed pretrain-adapt-finetune paradigm with longer adapt stage and fixed finetune stage to Table 5.

In the revised manuscript, we have marked the revisions in blue. We hope that our efforts address the reviewers' concerns.

---

### Meta-Review · Area_Chair_PvRE · 2022-09-06

**Recommendation:** Accept
**Confidence:** Certain

**Metareview:**


 This paper provides an empirical analysis of the effectiveness of self-supervised learning-based pre-training on multi-task learning, specifically for tasks within autonomous driving. After showing that the standard fine-tuning procedure does not work well in this context, the authors propose a pretrain-adapt-finetune procedure, involving multi-scale adapters and a language-to-vision adapter via task-specific prompt learning.

  The reviewers appreciated the focus on multi-task learning, especially in a domain where it is highly relevant and standard self-supervised learning methods are under-explored. The experiments were noted to be thorough, with results across many different tasks. However, several concerns were raised including a better set of ablations and adapter design exploration, and performance on another dataset. The author provided a thorough rebuttal, including significant new ablations/experiments and results on NuImages. All reviewers subsequently recommended acceptance, including the reviewer who had a borderline acceptance but mentioned the new results are convincing.

 While many of the elements of the proposed approach are not new, after the new experiments and ablations especially, this paper provides a nice contribution exploring an under-appreciated setting of multi-task learning and the use of pre-trained models for them. The experiments are thoroughly done and would be of value to the community. As a result, I recommend acceptance.

**Award:**

No

---

### Decision · Program_Chairs · 2022-09-14

Accept